# Proteomic Profile of Glyphosate-Resistant Soybean under Combined Herbicide and Drought Stress Conditions

**DOI:** 10.3390/plants10112381

**Published:** 2021-11-05

**Authors:** Rafael Fonseca Benevenuto, Caroline Bedin Zanatta, Miguel Pedro Guerra, Rubens Onofre Nodari, Sarah Z. Agapito-Tenfen

**Affiliations:** 1Crop Science Department, Federal University of Santa Catarina, Florianopolis 88034000, Brazil; rfbenevenuto@gmail.com (R.F.B.); carolzanatta77@yahoo.com (C.B.Z.); miguel.guerra@ufsc.br (M.P.G.); rubens.nodari@ufsc.br (R.O.N.); 2GenØk Centre for Biosafety, Siva Innovasjonssenter Postboks 6418, 9294 Tromsø, Norway

**Keywords:** energy cost budget, EPSPS, fitness cost, glyphosate, *Glycine max*, abiotic stress, transgenic

## Abstract

While some genetically modified (GM) plants have been targeted to confer tolerance to abiotic stressors, transgenes are impacted by abiotic stressors, causing adverse effects on plant physiology and yield. However, routine safety analyses do not assess the response of GM plants under different environmental stress conditions. In the context of climate change, the combination of abiotic stressors is a reality in agroecosystems. Therefore, the aim of this study was to analyze the metabolic cost by assessing the proteomic profiles of GM soybean varieties under glyphosate spraying and water deficit conditions compared to their non-transgenic conventional counterparts. We found evidence of cumulative adverse effects that resulted in the reduction of enzymes involved in carbohydrate metabolism, along with the expression of amino acids and nitrogen metabolic enzymes. Ribosomal metabolism was significantly enriched, particularly the protein families associated with ribosomal complexes L5 and L18. The interaction network map showed that the affected module representing the ribosome pathway interacts strongly with other important proteins, such as the chloro-plastic gamma ATP synthase subunit. Combined, these findings provide clear evidence for increasing the metabolic costs of GM soybean plants in response to the accumulation of stress factors. First, alterations in the ribosome pathway indicate that the GM plant itself carries a metabolic burden associated with the biosynthesis of proteins as effects of genetic transformation. GM plants also showed an imbalance in energy demand and production under controlled conditions, which was increased under drought conditions. Identifying the consequences of altered metabolism related to the interaction between plant transgene stress responses allows us to understand the possible effects on the ecology and evolution of plants in the medium and long term and the potential interactions with other organisms when these organisms are released in the environment.

## 1. Introduction

Currently, the global cultivated area of genetically modified (GM) crops has reached more than 190 million ha in a total of 29 countries [1]. The vast majority of such GM crops are engineered for herbicide tolerance or to produce their own insecticide. Herbicide-tolerant GM crops, which are able to accumulate herbicide residues without dying to facilitate weed management, occupy approximately 45% of the total genetically engineered crop cultivation area [1]. Of these, glyphosate tolerance traits represent the world’s most widespread transgenes [1,2]. This is due to the widespread use of glyphosate in global agriculture after the first commercial release of engineered glyphosate-resistant crops in 1996 [3], which has ever since caused global concerns regarding its potential environmental impact.

Herbicide application is considered one of the major abiotic stressors for plant metabolism [4]. Exposed transgenic plants have shown the activation of important stress response pathways to maintain basic physiological functions [4,5,6,7,8,9]. Other naturally occurring environmental stressors, such as drought, heat, and salt, can also negatively influence the crop’s ability to convert energy into biomass. Of these, drought is the most severe abiotic factor reducing crop productivity worldwide, and this has become more accentuated due to global warming [10]. Even more disturbing is the synergistic effect of the accumulation of various abiotic stressors, all of which negatively affect biomass production and consequently yield [11,12,13]. For instance, drought stress has been a recurring phenomenon in major growing regions where herbicide application is a common practice. There are concerns over GM crops, which under stress conditions and in combination with various genetic backgrounds can result in unexpected effects that pose risks for the environment and ecosystems [14]. Nevertheless, routine risk assessment analyses do not include information on the responses of GM plants exposed to different environmental stress conditions. Whereas changes in the fitness of GM plants are considered by, for instance, The European Food Safety Authority (EFSA), the assessment of their potential persistence and invasiveness in the ecosystem is restricted [14].

In our previous studies, the proteome profile of herbicide-tolerant maize, including its levels of phytohormones and related compounds, was analysed under drought and herbicide stresses. Twenty differentially expressed proteins and differences in abscisic acid (ABA), cinnamic acid (CA), jasmonic acid (JA), methyl ester of jasmonic acid (MeJA) and salicylic acid (SA) hormones were detected between GM and non-GM hybrids under different water deficiency conditions and herbicide sprays, mostly assigned to energetic/carbohydrate metabolic processes, as well as the plant’s response to biotic stressors. [15]. The findings supported the need for molecular and physiological studies of GM plants under variable environmental conditions during the risk assessment process. In another study, we were able to detect altered oxidative metabolism in herbicide-tolerant GM maize by altering the oxidative environment in cells, and the increased levels of antioxidant enzymes are likely to be a response to oxidative burst by reactive oxygen species (ROS) to maintain proper physiological function. In addition, glutathione metabolism was significantly altered in the plant when the herbicide Roundup was sprayed during cultivation [16,17]. Glutathione is known to be an important antioxidant in most living organisms, preventing damage to important cellular components caused by several environmental pollutants, including agrochemicals and plant glutathione S-transferases (GSTs), which are also widely known for their role in herbicide detoxification [9]. A cascade of enzymes involved in combating reactive oxygen species, such as ascorbate peroxidase, glutathione reductase, and catalase, were also expressed at a higher level in transgenic soybean seeds in another investigation [18]. Unintended effects of the inserted EPSPS-CP4 transgene were linked to energy metabolic disturbances in other studies [18,19,20]. It can be hypothesized that the plant is searching for a new equilibrium to maintain heterologous EPSPS-CP4 metabolism within levels that can be tolerated by the plant [16].

Glyphosate-tolerant GM plants need to overproduce the EPSPS enzyme with multiple copies of the *EPSPS* gene [21,22,23] and a strong promoter [24], attempting to provide sufficient EPSPS binding to glyphosate in order to reduce fatal toxicity. Studies have also shown that certain crops overexpressing *EPSPS* not only have increased glyphosate tolerance [2,24,25] but unexpectedly also show increased yield [26,27] and other fitness traits [28]. Such increases in fitness demand energy. Transgenic plants possessing traits for resistance to herbicides may display an ecological disadvantage compared to conventional plants in an herbicide-free environment due to fitness penalties [4]. Physiological fitness costs are generally mediated by pleiotropic effects on plant physiology, which can affect resource partitioning, growth rate, photosynthesis, phenology, seed germination and dormancy, and tolerance to abiotic factors [4]. Moreover, some environmental conditions may trigger the maximum expression of such fitness costs, while others may mask and make them negligible [29].

When considering the environmental risk of abiotic stress tolerance genes, it is important to take into account the magnitude of stress tolerance, the plant phenotype and the characteristics of the potential receiving environment [30]. In general, the current criteria used to declare a GMO substantially equivalent to its non-GM near-isogenic line are reduced to a limited set of predetermined compositional variables. Currently, risk assessment analyses do not include information on the response of GM plants exposed to adverse conditions. During times of climate change, drought periods have been increasingly recorded in major producing areas where herbicide application is an ordinary practice. Thus, it seems reasonable to assess the response of GM plants to variable environmental conditions, and nontargeted profiling tools based on omics technologies may help fulfil this gap. Proteomics profiling tools applied to the molecular characterization step of new GM plants under variable environmental conditions can assist in the identification of potentially harmful products (i.e., toxins, allergens, antinutrients). Therefore, aiming to provide insights into the metabolism of transgenic plants under accumulated abiotic stress conditions, we performed a nontargeted proteomics profiling analysis of herbicide-tolerant GM soybean under combined drought and herbicide stress conditions.

## 2. Materials & Methods

### 2.1. Plant Material and Growth Conditions

Four soybean varieties (*Glycine max*), two conventional BRS 283 and BRS 284 (near-isogenic controls), and two genetically modified (GM) varieties were used in this study. Both BRS 1001 and BRS 1010 iPRO, with the unique identifier MON-877Ø1-2 × MON-89788–1 of Monsanto Brasil SA, have resistance to glyphosate-based herbicide (GBH) and tolerance to some lepidopters due to the insertion of the recombinant protein Cry (rCry).

The experiment was carried out in a greenhouse in five factorial blocks with randomized treatments, with two lateral blocks serving as a border. Before sowing, all seeds were treated with the fungicide Derosal^®^ Plus (Carboxin + Thiram), with the inoculant Masterfix^®^
*Bradyrhizobium japonicum* and with cobalt (Co) and molybdenum microelements (Mo). Three plants per pot were grown at 15 cm spacing, with pots distributed between rows at 45 cm [31]. The 14 L pot [32] was filled with a substrate (1/3 clay soil; 1/3 cellulose residue, and 1/3 organic poultry residue) with the pH corrected to 6.0. One irrigation system was associated with each pot where the water conditions were maintained according to the field’s soil capacity [33,34].

In the V2 phenological stage, 22 days after sowing (DAS), the varieties BRS 1001 and BRS 1010 iPRO, referring to the treatment of stress by herbicide (represented by three plants per pot in 3 blocks), were sprayed with the Roundup Transorb^®^ formulation (Monsanto do Brasil S.A.) (Potassium salt of N-(phosphono-methyl) glycine (588 g/L; N-(phosphono-methyl) glycine acid (480 g/L; other ingredients (820 g/L)). Spray application occurred at a dosage of 2.77 L/ha, estimated through the average considered for each species of spontaneous predicted by the manufacturer in the product brochure. For each individual plant, a volume of 0.7 mL of herbicide was provided. This estimate considered a population of 280,000 soybean plants in 1 hectare and the flow rate of the manual sprayer. Six hours after application [35], three leaves were collected from the three plants present in the pot, immediately frozen in liquid nitrogen and stored at −80 °C.

In phenological stage V4, 39 (DAS), the plants corresponding to the drought stress treatments had their water supply interrupted (Figure 1). A VH400 Vegetronix sensor was used for the daily collection of soil electromagnetic conductivity. After seven days of treatment (45 DAS), the soil tension was below 0.5 MPa of pressure, and the moisture was close to <20% according to the retention curve [36]. At that moment, the leaves were collected and properly stored at −80 °C until protein extraction, as previously mentioned.

Each treatment was composed of a pool of nine samples from each variety. Thus, samples used for the proteomic analysis represented a total of 18 samples for each treatment: (1) GM varieties BRS 1001 and 1010, without herbicide application and drought stress (hereafter named ‘GM’); (2) GM varieties BRS 1001 and 1010, without herbicide application, with drought stress (hereafter named ‘GM_D’); (3) GM varieties BRS 1001 and 1010, with herbicide application, without drought stress (hereafter named ‘GM_H’); (4) GM varieties BRS 1001 and 1010, with herbicide application and drought stress (hereafter named ‘GM_H_D’); (5) Conventional varieties BRS 283 and 284, without herbicide application and drought stress (hereafter named ‘non-GM’; (6) Conventional varieties BRS 283 and 284, without herbicide application, with drought stress (hereafter named ‘non-GM_D’).

### 2.2. Sample Preparation for Proteomic Analysis

Total protein extraction was conducted following the methodology proposed by Carpentier et al. [37] with modifications. In summary, 100 mg of leaf samples was ground with lysis buffer (5 mM EDTA, 100 mM KCl, sucrose 30%, TRIS HCl pH 8.5 at 50 mM and protease inhibitor) using an automatic bead-beating homogenizer (Precellys^®^). Samples were then precipitated using methanol/ammonium acetate (100 mM). After precipitation, the samples were centrifuged, the supernatant was discarded, and the remaining precipitate was washed twice with cold 20% dithiothreitol (DTT) in acetone. Pellets were resuspended and dissolved in a buffer containing 1.5% urea and 100 mM tetraethylammonium bromide (TEAB) (pH 8.0). Protein quantification was performed on a spectrophotometer (reading at 562 nm) with the BCA protein kit (Novagen working reagent) and adjusted to a concentration of 2 µg/µL in total. Protein extracts were reduced with 10 mM DTT and alkylated with 20 mM iodoacetamide (IAA) at room temperature in the dark. For trypsin digestion, protein samples were diluted in 100 mM TEAB to reduce the urea concentration to less than 2 M, and trypsin was added at a 1:50 (*w*/*w*) trypsin-to-protein ratio for 18 h of digestion.

### 2.3. TMT Labelling and LC–MS/MS Analysis

Tandem Mass Tag (TMT) was used to label the digested protein peptides according to the manufacturer’s protocol for the 10-plex TMT kit (Thermo Fisher Scientific, Waltham, MA, USA). Of the 10 labels included in each kit, 6 were used per biological pool from each treatment. Labelled peptides were then purified and separated into 10 fractions by strong cation exchange (SCX) across an increasing salt concentration. The eluted peptide fractions were purified and extracted once again before being lyophilized for direct analysis by liquid chromatography-tandem mass spectrometry (LC–MS/MS).

Fractionated samples were resuspended in 100 μL of 50 mM ammonium bicarbonate, and 10 μL of each of the 10 fractions was loaded onto a 50-cm EASY-spray column (Thermo Fisher Scientific). Quantitative analysis was performed using an Orbitrap Velos-Pro mass spectrometer (Thermo Fisher Scientific) in positive ion mode. The peptides were separated by gradient elution from 5–80% 0.1% trifluoroacetic acid in acetonitrile (5–40% from 0 to 100 min, 40–80% from 100 to 110 min) at a flow rate of 300 nL/min. Mass spectra (*m*/*z*) ranging from 400 to 1600 Da were acquired at a resolution of 60,000, and the 10 most intense ions were subjected to MS/MS by HCD fragmentation with 35% collision energy.

### 2.4. Protein Identification and Quantification

The obtained raw tandem mass spectrometry (MS/MS) data were processed using Mascot’s Proteome Discoverer 2.3 (Thermo Fisher Scientific) to search the library for protein identification. The UniProt database was used, and the library search used Mascot version 2.3. Quantification was performed using MaxQuant software with Perseus (Max-Planck-Institute of Biochemistry). The precursor mass tolerance for the searches was set at 20 ppm, and the fragment mass tolerance was set at 0.8 ppm. The taxonomy selected was *Glycine max,* and three enzymatic mis-cleavages were allowed. Dynamic modifications selected in the search were Oxidation/+15.995 Da (M) and deamidated/+0.984 (N, Q), and static modifications were carbamidomethyl/+57.021 Da (C), TMT10plex/229.163 Da (K), and TMT10plex/229.163 Da (Any N-terminus). Only peptides with TMT reporter ion signal intensities for all samples were used for further bioinformatics analysis. Prior to statistical analysis, peak intensities were normalized for internal reference scaling (IRS) to correct the random variation occurring between TMT runs [38], followed by the sample loading (SL) and trimmed mean of M value (TMM) normalization methods (Robinson & Oshlack, 2010) to correct compositional bias between TMT experiments.

### 2.5. Statistical Analysis

The data were first normalized to the median distribution, auto-scaled, and log transformed to facilitate further comparative proteomic analysis. For interpretation of the numerical values, means and differences of means on the logarithmic scale were back-transformed to geometric means and ratios of geometric means on the original scale. Plotting results of exploratory analysis, such as hierarchical clustering and partial least square-discriminant analysis (PLS-DA), were performed by the R-program-based server MetaboAnalyst 5.0 [39]. Pairwise *t*-tests (*p* < 0.05) were performed for comparative proteomics analysis of the comparatives of interest in relation to the research questions. To estimate the genotype factor effect, the following pairwise comparisons were used: GM vs. non-GM; GM_D vs. non-GM_D. Similarly, the drought factor effect was estimated by comparing GM_D vs. GM and GM_H_D vs. GM_H. The herbicide factor effect was based on the GM_H vs. GM comparison. In addition, to explore the accumulated effect of genotype, drought, and herbicide factors, the following comparison was performed: GM_H_D vs. non-GM. The resulting *p* values were adjusted for false discovery rate (FDR) with the Bonferroni–Holm method (*p*-adj FDR < 0.05) to avoid false-positives [40]. Fold changes are presented in logarithm base 2 (Log2FC), a widely used transformation for a continuous spectrum of values, representing up- (positive) and down-regulated (negative) compound values in a reader friendly fashion. We considered those proteins as differentially abundant which only reached at least 1 Log2FC and *t*-test *p*-adj FDR < 0.05 for the respective comparison. Functional annotation and identification of enriched metabolic pathways were performed using the UniProt database (https://www.UniProt.org/, accessed on 3 November 2021) and KEGG pathway enrichment analysis (Kyoto Encyclopedia of Genes and Genomes) using the differentially abundant proteins and metabolites as input. Pathways with *p*-adj FDR < 0.05 were considered significantly enriched. Additionally, the Stitch database was used to produce biological networks of protein-metabolite interactions to facilitate data interpretation. Data uploaded into Stitch platform was based on the outcomes of the ANOVA comparative analyses (Appendix A). A cut-off score for the confidence of interaction ≥ 0.4 was used for a more reliable biological network.

## 3. Results

### 3.1. Exploratory Data Analysis

We analysed the leaf proteomic profile of herbicide-tolerant GM soybean and non-GM near-isogenic varieties under the same experimental conditions, optimal and herbicide/drought stress (Figure 2), to investigate the possible metabolic costs of GM plants under combined stress scenarios. TMT-based proteomic analysis identified 5894 proteins in our sample set. First, we performed an exploratory analysis of expressed proteins based on a supervised approach of partial least square-discriminant analysis modelling (PLS-DA) to explore all sources of variation in the dataset. All proteins were used for PLS-DA modelling, and a score plot was generated. By reducing the dimensionality of the data, the score plot of the first two components produced clusters grouping biological replicates according to the different treatments based on protein expression observations. The first component of PLS-DA shows a clear separation of the samples according to the ‘drought’ factor, accounting for 9.8% of the total variation. The second component of the model, representing 6.5% of the variation, separated the GM from the non-GM samples (‘genotype’ factor). The combination of the first two factors accounted for 16.3% of the total variation in the proteomic dataset (Figure 3A).

Second, we performed hierarchical clustering from the analysis of the top 50 protein hits (FDR corrected *t*-test *p* value < 0.05) (Figure 3B). The resulting heatmap showed similar results with two major groups of drought-stressed and control plants, followed by subgroups of GM and non-GM varieties. For both exploratory analyses, the ‘herbicide’ factor did not show clear separation in the dataset. This might be explained by the single application time point, which was done one month before proteomic sampling.

Overall, exploratory analysis results show that the main source of variation originated from the drought stress treatment, followed by the different soybean genotypes (GM and non-GM).

### 3.2. Metabolic Effects from Genetic Transformation

To assess any metabolic disturbances resulting from genetic transformation in the soybean, we conducted a pairwise statistical comparison between GM and non-GM plants under controlled environmental conditions (herbicide and drought). Although the transgenic variety expressed only two heterologous proteins from the stacked transgenic cassettes, both comparisons revealed 31 differentially expressed (DE) proteins between the GM and non-GM varieties. Under optimal conditions, the comparison between the GM vs. non-GM proteome profiles revealed eight significantly altered proteins, with five being upregulated and three downregulated in the GM variety. Under drought stress conditions (GM_D vs. non-GM_D), 23 proteins were differentially expressed between treatments, with eight upregulated and 15 downregulated in the GM plant (Appendix A).

We performed Stitch interaction network analysis to obtain insights into the molecular and cellular functions and metabolic pathway interactions of DE proteins and other small molecules. KEGG pathway enrichment analysis showed the ribosome pathway (FDR adj. *p* value = 7.6 × 10^−5^) as the most significantly altered pathway (36 protein ID hits vs. 41 protein ID entry). According to PFAM and INTERPRO domain enrichment, the enriched proteins in the pathway are related to ribosomal L5 and L18 family proteins. An interaction network map showed that the affected module representing the ribosome pathway strongly interacts with other important altered proteins, such as chloro-plastic ATP synthase gamma subunit (I1MFH3-GLYMA15G11490.1 ATPC, log2FC = −2.7); two ATP-binding proteins (I1LL18-GLYMA11G20040.1 and K7LTN0-GLYMA12G08430.2, log2FC = 2.8); a peptidylprolyl isomerase (C6TFQ0-GLYMA08G09480.1, log2FC = −3.1); and a protein disulfide-isomerase (A0A445H9T4-GLYMA14G40490.1, log2FC = −3.0) (Figure 4; Appendix A).

Stitch chemical prediction showed MgADP, glucose, phosphoenolpyruvate (PEP), and carbon dioxide as interaction partners to the DE proteins. The involved biochemical pathways are related to carbon fixation, carbohydrate metabolism (beta galactosidase enzyme (GLYMA13G42560.1; log2FC = −3.0) and the phosphoenolpyruvate carboxylase enzyme (GLYMA13G36670.1; log2FC = −3.1)). It also includes the metabolism of N-glycan processing and the metabolism of protein folding in the endoplasmic reticulum (glucosidase) (GLYMA07G35090.1; –log2FC = −2.9); the calreticulin pathway (GLYMA09G38410.2; log2FC = −3.0]); redox metabolism (aldehyde dehydrogenase domain-containing protein) (GLYMA09G32170.1; log2FC = −2.9) and protein-methionine-S-oxide reductase metabolism (GLYMA04G36480.1; log2FC = −2.5))). All these proteins were shown to be downregulated in the GM variety.

### 3.3. Metabolic Effects of Combined Herbicide and Drought Conditions

To understand the impact of combined abiotic stressors, we compared the proteomic profiles of GM and non-GM samples when stressors were applied alone or in combination.

First, GM plants treated with glyphosate-based herbicide were compared to the control GM samples (GM_H vs. GM). A total of 21 proteins were significantly affected in response to herbicide application, with nine being upregulated and 12 downregulated, and 18 protein ID hits identified by Stitch program. Similar to previous results, KEGG pathway enrichment analysis also revealed the ribosome pathway as the most significantly enriched pathway (FDR adj. *p* value = 0.014). These proteins can be observed as red dots in the metabolic network shown in Figure 4. PFAM and INTERPRO analysis assigned these proteins to ribosomal protein S9 and S5 domains. The Stitch network also predicted significant interactions with MgATP, guanosine triphosphate, uridine triphosphate, and cytidine (araCTP) chemicals. MgATP strongly interacted with magnesium chelatase (I1LAA1–GLYMA18G43240.2, log2FC = −2.9), a downregulated protein involved in chlorophyll biosynthesis, as well as the acetyl-CoA and fatty acid biosynthesis proteins beta-ketoacyl synthase I (Q9M507-GLYMA05G25970.1 LOC548028, log2FC = −3.2) and uncharacterized acetyl-CoA dehydrogenase protein (K7MTK7-GLYMA18G43240.2, log2FC = −2.7). Protein involved in transcription was uncharacterized DNA-binding protein (C6SXZ4–GLYMA09G32550.1, log2FC = −3.1) and in dephosphorylation was purple acid phosphatase PAP3 (Q6YGT9-GLYMA17G11790.1 PAP3, log2FC = −3.0), which was also shown to play a central role in the altered metabolic network (Figure 5; Appendix A). In addition, herbicide treatment resulted in significant overexpression of proteins linked to the ABA signaling pathway and abiotic stress response: C2 domain-containing protein (A0A0R0I760–not in the map, log2FC = 3.3) and uncharacterized remorin-like protein (C6SVM4–not in the map, log2FC = 2.7); cell redox homeostasis: thioredoxin domain-containing protein (I1KDX9–GLYMA06G37970.1, log2FC = 2.9); ATP and peptide hydrolysis: Obg-like ATPase (K7L8F3-GLYMA08G25673.1, log2FC = 3.1) and uncharacterized peptidase protein (I1MHG0–GLYMA15G19580.1, log2FC = 3.1).

Drought stress effects on GM soybean metabolism were assessed by comparing the proteome profiles of GM_D vs. GM, as well as in accumulation with herbicide stress effects through the GM_H_D vs. GM_H comparison. In total, 117 proteins were differentially expressed between the treatments, with 64 upregulated and 53 downregulated in response to drought. The KEGG pathway enrichment analysis network revealed that most DE proteins were assigned to important stress-related pathways.

To facilitate the interpretation of the data, we highlighted five major functional modules that correspond to the most significantly enriched pathways (257 protein ID hits in Stitch versus 257 protein IDs entry) (Figure 6; Appendix A). As expected, biosynthesis of secondary metabolites (FDR adj. *p* value = 0.003) was among the altered pathways, with DE proteins also involved in other metabolic modules (red dots in the map). The first module includes the sulfur (FDR adj. *p* value = 3.5 × 10^−5^) and purine (FDR adj. *p* value = 0.0002) metabolic pathways. These altered pathways showed a strong interaction with MgADP, a significant predicted chemical playing a central role in the interaction network of DE proteins. Module 2 is composed of altered pathways of alpha-linoleic metabolism (FDR adj. *p* value = 0.003), biosynthesis of unsaturated fatty acids (FDR adj. *p* value = 0.004), as well as metabolism (FDR adj. *p* value = 0.04) and degradation (FDR adj. *p* value = 0.019) of fatty acids. Most of the DE proteins associated with these pathways were downregulated in response to drought stress (K7L1W0-GLYMA07G18370.2; A0A0R0F201-GLYMA18G43240.2; A0A0R0KRB3-GLYMA03G07540.1; log2FC = −2.7), but one associated with steroid metabolism was altered under combined herbicide and drought stresses (I1LJJ4-CYP74A2, log2FC = −2.7). Monoterpenoid biosynthesis pathway (FDR adj. *p* value = 0.012), module 3 in the network, was upregulated in the drought-stressed plants. On the other hand, module 4 represented the glutathione metabolic pathway (FDR adj. *p* value = 0.001), which included the DE proteins glucose-6-phosphate 1-dehydrogenase (A0A0R0FM59-GLYMA16G06850.1, log2FC = −2.5) and glutathione transferases (K7KBB0-GLYMA02G45336.1, log2FC = −2.7; Q9FQD4-LOC547951, log2FC = −2.7; I1KK55-GLYMA07G16800.1, log2FC = −2.2; K7L1L7-LOC547577, log2FC = −2.2), which were significantly downregulated in GM plants under accumulated herbicide and drought stresses. Additionally, module 5 represents the altered pathways involved in genetic information processes, ribosome (FDR adj. *p* value = 0.018) and protein export (FDR adj. *p* value = 0.018).

### 3.4. Cumulative Effects of Genetic Transformation and Abiotic Stress Conditions

We next evaluated the metabolic effects on the combination of the genetic transformation process under cumulative herbicide and drought stress scenarios. We compared the proteome profiles of GM soybean samples under combined herbicide and drought applications to the unmodified control under optimal conditions (GM_H_D vs. non-GM). In total, 126 proteins were differentially regulated, whereas 66 were upregulated and 60 were downregulated in the transgenic variety. The interpretation of the network interaction map was facilitated by limiting the inclusion of pathways with high-confidence interaction scores only (>0.700). Glutamic acid, ammonia, NADPH, and ketoglutarate were added as predicted significant chemicals interacting with the groups of the altered proteins found. Pathway enrichment analysis of DE proteins showed biosynthesis of secondary metabolites (FDR adj. *p* value = 0.006) as one of the most altered pathways, with 14 DE proteins enriched (red dots in the interaction network). The interaction network was divided into five functional modules according to the results of KEGG pathway enrichment analysis (161 protein ID hits on Stitch versus 231 protein IDs entry) (Figure 7; Appendix A).

Module 1 shows the ribosome pathway (FDR adj. *p* value = 7.3 × 10^−7^, which has 14 protein hits and has been shown to be affected in all conditions in our dataset.

Module 2 represents sulfur (FDR adj. *p* value = 7.0 × 10^−7^) and purine (FDR adj. *p* value = 0.002) metabolic pathways, which were also affected when drought conditions were applied. This functional module showed interaction with module 3, which was composed of pathways involved in carbon metabolism (FDR adj. *p* value = 0.023) and fixation (FDR adj. *p* value = 0.046), folate metabolism (FDR adj. *p* value = 0.046), as well as amino acid biosynthesis (FDR adj. *p* value = 0.023) and metabolism (FDR adj. *p* value = 0.026). The interaction between Module 2 and Module 3 is mediated by the predicted chemicals NADPH and ketoglutarate.

Module 3 revealed key proteins involved in the main steps of carbon and energy metabolism, which were significantly downregulated in the GM-treated plants. These proteins are fructose-bisphosphate aldolase (I1MB71–GLYMA14G36850.1, log2FC = −2.9), which is involved in the main steps of glycolysis, and TCA cycle enzymes of malate dehydrogenase (Q9SPB8–GLYMA12G19520.1 MDH-1, log2FC = −2.9) and succinate–CoA ligase (I1MQH3–GLYMA16G33870.1, log2FC = −2.9), as well as the tetrahydrofolate reaction proteins of one-carbon metabolism, serine hydroxy-methyl-transferase (A0A0R4J2Q8-GLYMA02G38160.3, log2FC = −2.7) and formyltetrahydrofolate synthetase (I1NJ85-GLYMA20G38740.1, log2FC = −2.9). Interestingly, the overexpression of some enzymes involved in amino acid and nitrogen metabolism, such as glutamate synthase (A0A0R0KBX5–GLYMA04G41540.1, log2FC = 2.2), phospho-2-dehydro-3-deoxyheptonate aldolase (I1JGU8–GLYMA02G37080.1, log2FC = 2.5), phosphoglycerate kinase (A0A368UH36–GLYMA15G41550.1, log2FC = 1.6), and N-acyl-L-amino-acid amidohydrolase (I1LW56–GLYMA13G02870.1, log2FC = 1.6), are related to the altered energy budget seen in Module 1 in GM-treated plants.

Module 4 represents the porphyrin and chlorophyll metabolic pathways (FDR adj. *p* value = 0.026), which showed a lower abundance of ferritin proteins (I1MYZ8-GLYMA18G02800.1 ferritin-3, log2FC = −2.4; I1JIE0-GLYMA02G43040.1 Fer2-1, log2FC = −1.6) involved in redox homeostasis of chloroplasts and Fe storage and bioavailability.

Last, Module 5 encompasses galactose metabolism (FDR adj. *p* value = 0.041), showing significant downregulation of aldose 1-epimerase (I1N4B8-GLYMA18G49210.1, log2FC = −2.8) and the cell wall hydrolase enzyme beta-galactosidase (I1M5B6-GLYMA13G42560.1, log2FC = −2.6), as well as upregulation of the invertase enzyme beta-fructo-furanosidase (I1MUX8-GLYMA17G14750.1, log2FC = 1.1).

## 4. Discussion

### 4.1. Metabolic Costs Inferred by Pleiotropic Effects in Genetically Modified Soybeans

Pleiotropic effects of herbicide-resistance genes have been shown to be harmful to fitness and yield and have been linked to resistance mechanisms that originate from genetic resources, mutagenesis and genetic engineering in crops [41]. On the other hand, such effects of resistance genes were not always well distinguished from other sources of variation. Over the last few years, omics techniques have been successfully used as valuable tools for the investigation of unintended metabolic changes in transgenic crops under variable environmental conditions [15,16,17,42,43,44,45]. The successful application of omics, such as proteomics, in the molecular characterization of GM plants has been demonstrated in recent studies, particularly because the plant proteome profile directly reveals the physiological state under different environmental conditions. In the current work, we found evidence of metabolic burden in the regulation of ribosomal proteins and the alteration of carbohydrate and energy metabolism in response to cumulative abiotic stresses in stacked GM soybean varieties.

In this study, we investigated the unintended changes in the metabolism of stacked insecticide and herbicide-tolerant GM soybean due to potential pleiotropic effects. We observed persistence alterations in the ribosome pathway of the GM soybean proteome profile under controlled and accumulated herbicide- and drought-stressed conditions. The majority of ribosome pathway proteins were identified as chloro-plastic ribosomal proteins L5 and L18 from the 60S and 50S subunits, as well as directly interacting proteins involved in isomerase activity, chaperone protein folding, and ATP synthesis. The ribosome is an essential ribonucleoprotein complex that is mainly involved in translation and indispensable for plant growth and development. Ribosomal proteins are well known for their universal roles in forming and stabilizing the ribosomal complex and mediating protein biosynthesis, a highly energy-demanding process. Any deregulation in such complex ribosomal proteins, as well as extra-ribosomal proteins involved in translation, may directly cause limitations in translational activity, indirectly affecting normal plant growth and developmental processes [46].

In addition, translational regulation is dependent on some extra-ribosomal proteins. For instance, di-sulfide and peptidylprolyl proteins, two important proteins involved in the isomerase activity of chloroplasts and the endoplasmic reticulum (ER), were less abundant in the GM plant than in its non-GM isoline in response to drought stress. These are two protein-folding machineries, namely, foldases, which work together with chaperones to facilitate the folding of every protein by catalyzing isomerization of prolyl peptides or disulfide bonds for proper folding. Other ER-located protein folded mediators, such as calreticulin and glucosidases, were also downregulated in the GM plant compared to the non-GM variety. In general, plant stress is known to enhance the expression of ER chaperones and foldases [47,48,49]. The high demand for protein folding causes ER stress and activates signaling cascades, leading to an enhancement in protein folding capacity [50].

Remarkably, under optimal growing conditions, the GM soybean variety presented deficiency in the abundance of beta-galactosidase, an important cell wall hydrolase that is likely to play key roles in osmotic adjustments in leaves during abiotic stress, including drought [51,52]. Nevertheless, these enzymes participate in the breakdown of cell wall polysaccharides, which have been appointed as part of the catabolic network that provides respiratory sugars when other cellular sources for sugar production are exhausted [53,54]. Moreover, three important enzymes involved in ATP synthesis and binding, the main cell source and use of energy and phosphate, also appeared to be altered. ATPases involved in ATP binding and transport of the energy provided from ATP breakdown to perform other cellular reactions were overexpressed in the GM variety under optimal growing conditions.

Enzymes involved in redox activity (oxidoreductases) were also regulated differently in genetically engineered soybean plants. For instance, aldehyde dehydrogenase, a family of enzymes working on aldehyde homeostasis contributing to redox balance [55], was shown to be less expressed in the transgenic variety under controlled growing conditions.

These findings provide evidence for the existence of a metabolic burden, particularly with regard to energy and protein-related metabolism of GM plants, probably as a physiological penalty associated with metabolic drains resulting from genetic transformation. Additionally, the GM soybean variety appears to have a more costly response to drought stress stimuli than its non-GM near-isogenic line. The constitutive expression of the inserted transgenes is often controlled by strong viral promoters, which can be problematic, as the constitutive expression of heterologous proteins may compete for energy, resulting in energetic costs to the plant’s metabolism and consequent growth and development. In their review paper, Singhal et al. [56] showed that constitutive expression of a variety of transgenes causes either abnormal growth under normal conditions or mild to severe growth retardation in the aerial parts and reduced sugar content compared to non-transformed plants.

Intriguingly, when we analysed the isolated effects of GBH on GM soybean metabolism, in addition to the ribosome pathway, in our biological network analysis, we did not find additional pathways that were significantly affected. In contrast, previous studies have found a significant imbalance in carbohydrate and energy metabolism of GM plants due to GBH application, as well as changes in glutathione metabolism as an indication of increased oxidative stress [15,16,17]. Other studies have also found GBH effects on photosynthesis and nutrient accumulation in glyphosate-resistant soybean [8,9,57]. However, such studies that analysed soybean leaf material have been sampled from 8 to 24 h after glyphosate treatment [15,17]. In the current study design, GM plants were treated with GBH in V2 (approximately 34 DAE), but leaf sampling was conducted together with the other treatments (i.e., drought stress and accumulated herbicide + drought stress) when plants were between V5 and V6, which was 13 days after herbicide application. This suggests that the GM plants have a window of sensitivity to GBH treatment. In this study, plants were treated with herbicide just once and approximately one month before sampling. Glyphosate-tolerant plants appear to undergo a peak of oxidative stress immediately after GBH application as a cellular detoxification process and consequent alteration of energy metabolism to resist harmful effects. After this stress response period, the plant’s metabolism needs some time to regenerate to survive and successfully complete its plant life cycle. Zobiole et al. [57] found that glyphosate application at later growth stages decreased nutrient accumulation, nodulation and the parameters of plant growth and yield, making the plant more susceptible in the last physiological stages. This is additional evidence for the sensitivity window of tolerant GM soybean as an immediate response to the herbicide, a period in which the plants appear to be more vulnerable [58].

### 4.2. Metabolic Costs from Combined Abiotic Stressors

The accumulation of abiotic stresses, such as herbicide application and drought periods, during the plant life cycle is a reality. By investigating changes in the proteome of GM plants under herbicide plus drought stress, we found an increased metabolic alteration in response to cumulative abiotic stresses. In addition to ribosome and protein export metabolism, GM plants activate sulfur and purine metabolism, fatty acid biosynthesis and metabolism, biosynthesis of monoterpenoids, and glutathione metabolism. Most of these pathways have already been described in previous studies as being altered in response to drought and herbicide stress in GM plants [15,17,18,19,20]. However, when adding the effect of genetic transformation to herbicide and drought stress factors, the present study detected the alteration of additional pathways, suggesting an accumulation of metabolic costs. For instance, additional changes in the regulation of pathways related to carbon and amino acid metabolism, as well as galactose, porphyrin and chlorophyll metabolism, were found to be cumulative metabolic burdens in the GM plants under combined abiotic stresses.

The downregulation of key enzymes involved in the main steps of carbohydrate metabolism (i.e., glycolysis and TCA cycle), together with overexpression of enzymes from amino acid and nitrogen metabolism, may be related to a more complex imbalance in the carbohydrate and energetic metabolism of GM-stressed plants. Previous studies that analysed the isolated effects of herbicide and drought in the proteome of single and stacked GM plants have described only upregulation of such enzymes involved in glycolysis- and carbon fixation-related pathways [15,17]. In addition, significant changes in the regulation of epimerases, invertases, and beta-galactosidase involved in galactose metabolism also revealed additional alterations in further steps of sugar metabolism in the cell wall because of accumulated genetic modification and combined abiotic stress effects. Although little is known about the metabolic roles of wall polysaccharides in sugar reprogramming, this has been linked to work in the background of loss in photosynthetic production of sugar-developing mechanisms of cellular reprogramming to sustain energy homeostasis when experiencing stress, through a complex catabolic network to provide respiratory sugars to energy-starved cells [54].

Alterations in porphyrin and chlorophyll metabolism of GM plants also showed a significantly lower abundance of ferritin proteins, a key protein that modulates reactions of redox homeostasis in chloroplasts and plays a central role in Fe storage and bioavailability [59,60]. Fe deficiency results in reduced activities where Fe is required, such as electron transport in photosynthesis and sulfur and nitrogen metabolism, which can compromise plant growth [61]. Fe deficiency also leads to the expression of reactive oxygen species (ROS)-scavenging to prevent ROS-induced damage, which accumulates due to the impairment of photosynthetic electron transport because of reduced Fe availability [62,63].

Additionally, the accumulation of antioxidant enzymes mainly occurred in the glutathione metabolism of GM-tolerant soybean in response to GBH treatment, as found in the present and previous studies [16,17,18]. The accumulation of genetic transformation stress due to constitutive expression of two transgenic cassettes, plus drought and herbicide exposure, indicated an even more costly oxidative stress to maintain cellular redox homeostasis. When considering the cytoplasm environment and the signal transduction routes in response to a range of stress stimuli activated in different domains of organelles, it is assumed that the steady state is homeostasis and that any alteration of it has costs. Under drought, the expression of the other four oxidoreductases significantly changed in the GM, suggesting a considerable effect of genetic transformation on responses to oxidative stress. Redox imbalance is known to damage cell integrity and can negatively interfere with photosynthetic processes by decreasing the chlorophyll content, leading to a reduction in plant growth [17].

Taking these findings together, we found clear evidence for increasing the metabolic costs of GM soybean plants in response to the accumulation of stress factors. First, alterations in the ribosome pathway indicate that the GM plant itself carries a metabolic burden associated with the biosynthesis of proteins as effects of genetic transformation. GM plants also showed an imbalance in energy demand and production under controlled conditions, which was empowered under drought conditions. Isolated herbicide stress revealed that GM plants have a window period of sensitivity responses immediately after the application, which may recuperate after such a vulnerable period. The combination of drought and herbicide stress resulted in metabolic alterations regarding ribosome, sulfur and purine metabolism, fatty acid metabolism, biosynthesis of monoterpenoids, and glutathione metabolism. When the effects of genetic transformation are combined with accumulated abiotic stresses, more complex changes related to carbohydrate and energy metabolism and redox homeostasis disturbances were detected. Such findings suggest a staggering complexity of metabolic alterations as a response to the accumulation of stress factors and, consequently, the higher the costs are to deal with this scenario.

### 4.3. Relevance of Abiotic Stress Testing for Risk Assessment of GM Plants

In this study, analysis of the plant proteome revealed metabolic penalties due to stacking Cry and EPSPS transgenic traits in soybean GM varieties. When a combination of herbicide and drought stresses was applied to GM stacked varieties, proteins related to cellular stress response, carbon metabolism, chloro-plastic proteins, and ribosomal and secondary compounds were pronounced.

Abiotic stresses are recognized as potential causes of imbalance in the metabolism of regulation between source sinks in transgenic plants [64,65]. Transgene expression is negatively affected when stress response mechanisms are in place. For example, GM wheat showed altered Pm3b transgene expression under fungicide spraying [66] and in white maize varieties exposed to cold and wet conditions where the Bt content increased by an order of four times [67].

The application of herbicides is recognized as an abiotic stressor even when sprayed on tolerant GM crops. Similar to the results of the present study, glyphosate-based herbicides have been associated with impaired metabolic relationships even in crops that express resistant EPSPS. It has been previously shown to have a phytotoxic effect on seeds [68] and to alter carbon flow and oxidation [17]. The expression of stress-related genes has been observed in early GM soybean seedlings sprayed with glyphosate [69]. The impact on agronomic performance was observed by the reduction in photosynthetic activity and nodular activity and biomass decrease [7,57,70,71]. In addition, there seems to be a greater susceptibility to pathogens due to the accumulation of macro- and micro-nutrients [57] and less lignin deposition [9].

Considering the alterations found in GM soybean under the effects of stressors, different metabolic pathways were compromised. These effects suggest a negative impact on plant composition, agronomic performance, and alterations in gene expression and regulation. GM crop environments must be monitored from a biosafety perspective to verify potential risks in relationships with other biological organisms [14,72]. Omics techniques, including proteomics, provide highly sensitive mapping to investigate risk possibilities to be assessed by regulatory agencies [15,16,17,42,44,73,74]. Identifying the consequences of altered metabolism related to the interaction between plants and stress allows us to understand the possible effects on the ecology and evolution of plants in the medium and long term and the potential interactions with other organisms when these organisms are released in the environment.

## Figures and Tables

**Figure 1 plants-10-02381-f001:**
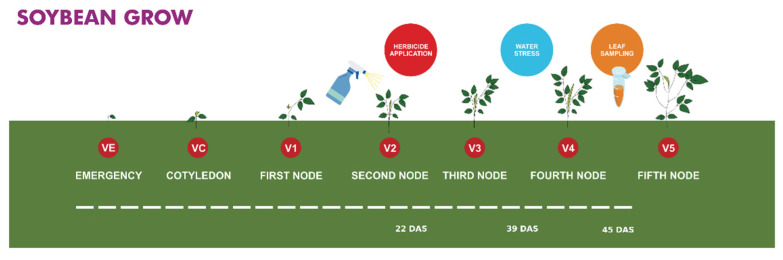
Schematic representation showing the timeline of soybean growth stages. Herbicide application, drought stress, and sample collection have been also included in color balloons indicating the time of the event within the soybean cultivation period.

**Figure 2 plants-10-02381-f002:**
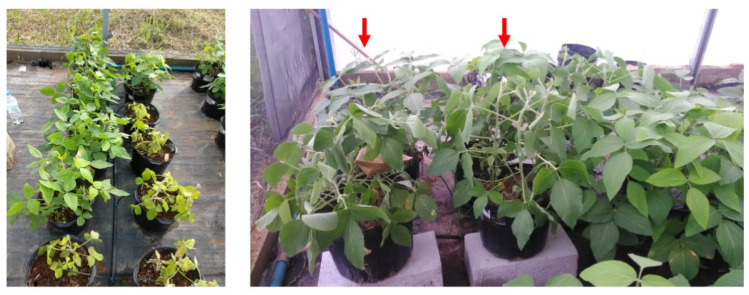
Soybean samples under herbicide and drought conditions. The left picture shows herbicide-treated (i.e. chlorosis symptoms) and untreated soybean plants (i.e. healthy plants). The right picture shows side-by-side soybean samples under drought stress at V5 stage with dehydration symptoms (red arrows) and control plants (optimal water conditions).

**Figure 3 plants-10-02381-f003:**
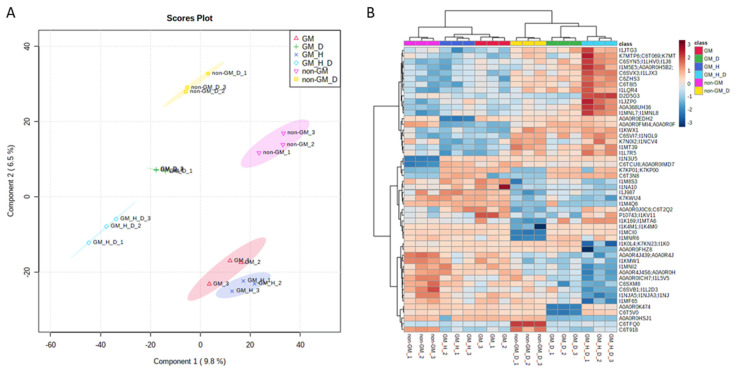
Exploratory analysis of proteomic profiles. (**A**) Scores plot for Partial least square-discriminant analysis (PLS-DA). Ovals show 95% confidence interval for each treatment. This is a scatter plot for the 2 components having the greatest variations. Observations that are similar will fall close to each other displaying a clustering-like pattern. Component 1 (*X*-axis) contains 9.8% of the total variation and component 2 contains 6.5%. (**B**) Hierarchical Clustering heatmap. Hierarchical clustering was performed considering the top 50 FDR corrected *t*-test *p* value (<0.05) passing proteins. Horizontal axis shows the biological samples analyzed in the study and vertical axis denotes Uniprot accessions for the proteins. On top of the heatmap samples are identified by treatment. Dendrogram for samples is shown on top of the heatmap and proteins’ dendrogram on left side of the heatmap. Dark blue to dark red color gradient denotes lower to higher expression.

**Figure 4 plants-10-02381-f004:**
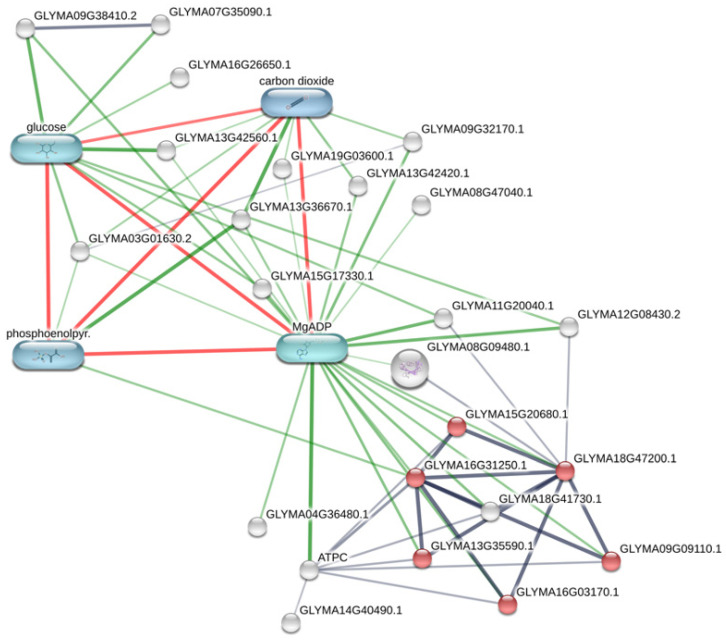
Biological network of differentially expressed proteins in response to genetic transformation of GM soybean plants. The visual network was built using Stitch database. Additional metabolites (green) were added as functional partners in order to show network context. Stronger associations are represented by thicker lines. Protein–protein interactions are shown in grey, chemical–protein interactions in green and interactions between chemicals in red. Image is automatically retrieved from the Stitch software. No manual entries have been made.

**Figure 5 plants-10-02381-f005:**
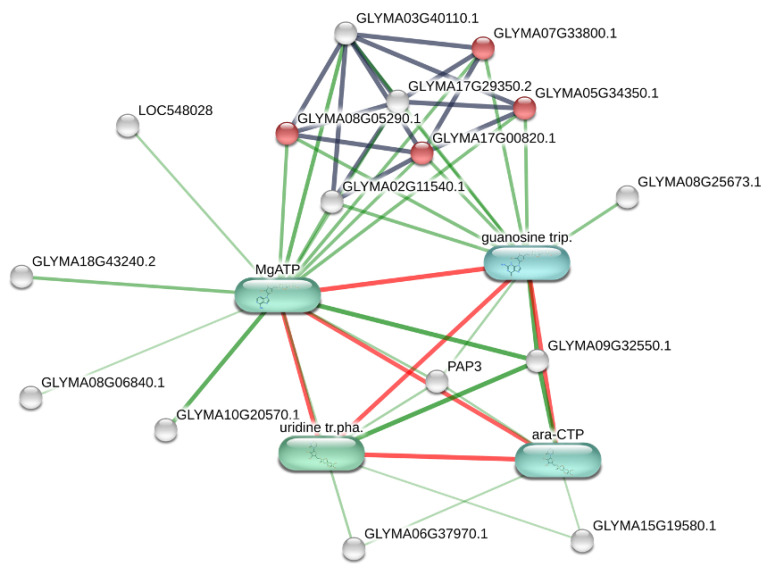
Biological network of differentially expressed proteins in response to herbicide application of GM soybean plants. The visual network was built using Stitch database. Additional metabolites (green) were added as functional partners in order to show network context. Stronger associations are represented by thicker lines. Protein–protein interactions are shown in grey, chemical–protein interactions in green and interactions between chemicals in red. Image is automatically retrieved from the Stitch software. No manual entries have been made.

**Figure 6 plants-10-02381-f006:**
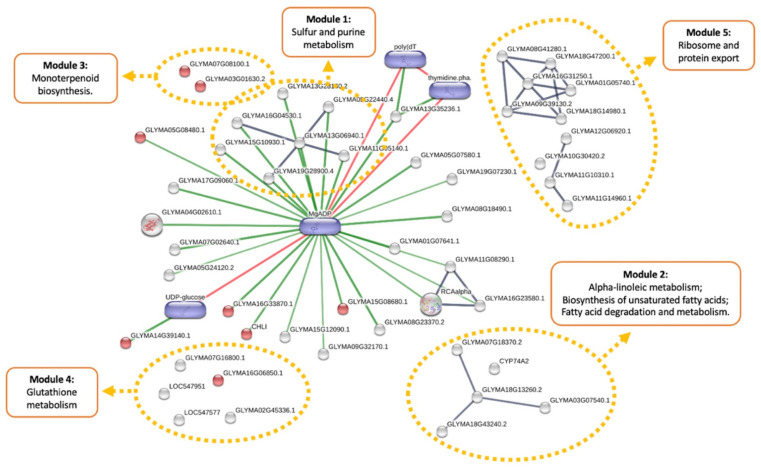
Biological network of differentially expressed proteins in response to combined herbicide and drought stress in GM soybean plants. The visual network was built using Stitch database. Functional modules are highlighted in orange dotted lines. Additional metabolites (purple) were added as functional partners in order to show network context. Stronger associations are represented by thicker lines. Protein-protein interactions are shown in grey, chemical–protein interactions in green and interactions between chemicals in red. Image is automatically retrieved from the Stitch software. No manual entries have been made.

**Figure 7 plants-10-02381-f007:**
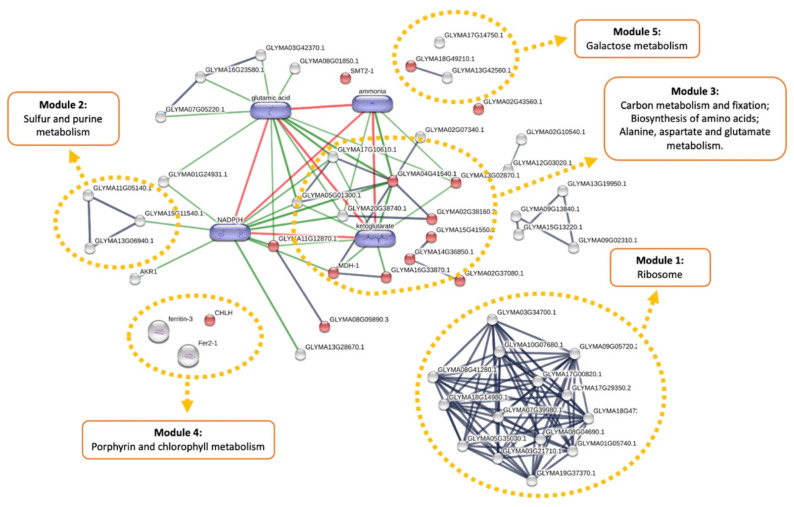
Biological network of differentially expressed proteins in response to accumulated effects of genetic transformation, herbicide and drought stress in soybean plants. The visual network was built using Stitch database. Functional modules are highlighted in orange dotted lines. Additional metabolites (purple) were added as functional partners in order to show network context. Stronger associations are represented by thicker lines. Protein–protein interactions are shown in grey, chemical–protein interactions in green and interactions between chemicals in red. Image is automatically retrieved from the Stitch software. No manual entries have been made.

## Data Availability

The data presented in this study are available in article and Appendix A.

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
