# Peer review of "Proteomic Profile of Glyphosate-Resistant Soybean under Combined Herbicide and Drought Stress Conditions"

_plants, 2021, doi:10.3390/plants10112381_

Round 1

Reviewer 1 Report

Manuscript titled "Proteomic profile of glyphosate-resistant soybean under
combined herbicide and drought stress conditions". It helps to understand the consequences of altered metabolism caused by a combination of interactions between plants, transgenes, and stress response enables us to understand the medium- and long-term effects on plant ecology and evolution. The manuscript is well written; therefore, it can be accepted for publication.

Author Response

Professional english revision has been performed by Springer-Nature Author Services. Please see certificate attached. We thank the referee for his/her evaluation.

Reviewer 2 Report

In my opinion, a well-designed, well-implemented and well-written report on an intriguing subject.  With the use of "omics" tools, the authors have investigated the joint and separate effects of herbicide spray and drought stress on GM soybean.  Perhaps we should not be surprised that substantial changes in a variety of metabolic pathways were uncovered.  Still, the specific changes, and the degree of those changes was remarkable.  Obviously, we need to carefully consider how our approaches to genetic modification may impact the adaptability of crop plants to changing environments.  Some editing for English language needed.

Author Response

We have sent the manuscript for professional English editing by Springer-Nature Author Services. We thank the referee for her/his comments and evaluation of the manuscript. Please see certificate in attach.

Reviewer 3 Report

The presented work is devoted to the study of protein metabolism of transgenic and non-transgenic soybeans under conditions of abiotic stress. The volume of the research done is large, the results are interesting and important. Studying the protein pool is a very complex and time consuming study. It's amazing how the authors were able to identify 5894 proteins in soybean samples by the TMS method. Even 117 proteins are difficult to identify. The results are discussed in detail in the Discussion section. However, the conclusions had to be made in the Conclusion section. The manuscript is well-formatted. There are several questions:
1. In Figure 1, show individual plants, as the bottom photo does not show the difference in the soybean samples.
2. In figure 3, remove the black background, the letters are not visible
3. How many have the authors been able to identify proteins with known functional and enzymatic activity?
4, Figures 4, 5 and 6 show the proteins identified in the databases with known activity?
5. How are data and coefficients entered into the Stitch program? Is it directly connected to mass spectrometry?

Author Response

Reviewer #3

The presented work is devoted to the study of protein metabolism of transgenic and non-transgenic soybeans under conditions of abiotic stress. The volume of the research done is large, the results are interesting and important. Studying the protein pool is a very complex and time consuming study. It's amazing how the authors were able to identify 5894 proteins in soybean samples by the TMS method. Even 117 proteins are difficult to identify. The results are discussed in detail in the Discussion section. However, the conclusions had to be made in the Conclusion section. The manuscript is well-formatted. There are several questions:

  1. In Figure 1, show individual plants, as the bottom photo does not show the difference in the soybean samples.
  2. In figure 3, remove the black background, the letters are not visible
  3. How many have the authors been able to identify proteins with known functional and enzymatic activity?

4, Figures 4, 5 and 6 show the proteins identified in the databases with known activity?

  1. How are data and coefficients entered into the Stitch program? Is it directly connected to mass spectrometry?

Response to reviewer’s #3 comments:

  1. The reviewer is probably talking about Figure #2. We have improved the Figure legend so that differences between soybean plants can be observed. We thank the reviewer for that suggestion.
  2. There is no black background in Figure 3. Perhaps this is a problem with incompatible software? We experienced no problems when opening the files using Microsoft Word Program version 16.54 for Mac.
  3. We have added the number of protein hits versus the number of uploaded proteins in Stitch for each analysis. This is indeed relevant data and we thank the referee for this question.
  4. Stitch provides biological networks based on protein annotations in verified databases. No manual annotation is added. The algorithm also includes metabolites present in the same pathway. The different colors in lines indicate that.
  5. Stitch analysis is not directly connected to the mass spec data. This is also because we performed our ANOVA analysis in R environment. So data from R was used to upload the protein networks into Stitch Program. This data can be retrieved from each of the four Supplementary tables. We have revised the text on this method and we hope this is clear now.